# Evaluation of a Wearable Non-Invasive Thermometer for Monitoring Ear Canal Temperature during Physically Demanding (Outdoor) Work

**DOI:** 10.3390/ijerph18094896

**Published:** 2021-05-04

**Authors:** Charlotte Christina Roossien, Audy Paul Hodselmans, Ronald Heus, Michiel Felix Reneman, Gijsbertus Jacob Verkerke

**Affiliations:** 1Department of Rehabilitation, University of Groningen, University Medical Center Groningen, Medicine, 9713 GZ Groningen, The Netherlands; m.f.reneman@umcg.nl (M.F.R.); g.j.verkerke@gmail.com (G.J.V.); 2Center for Applied Research and Innovation in Health Care and in Nursing, Hanze University of Applied Sciences, 9747 AS Groningen, The Netherlands; a.p.hodselmans@pl.hanze.nl; 3Institute for Safety (IFV), Knowledge Center Occupational Safety, 6816 RW Arnhem, The Netherlands; ronald.heus@ifv.nl; 4Department of Biomedical Engineering, University of Twente, 7522 NB Enschede, The Netherlands

**Keywords:** thermal physiology, heat strain, overheating

## Abstract

Aimed at preventing heat strain, health problems, and absenteeism among workers with physically demanding occupations, a continuous, accurate, non-invasive measuring system may help such workers monitor their body (core) temperature. The aim of this study is to evaluate the accuracy and explore the usability of the wearable non-invasive Cosinuss° °Temp thermometer. Ear canal temperature was monitored in 49 workers in real-life working conditions. After individual correction, the results of the laboratory and field study revealed high correlations compared to ear canal infrared thermometry for hospital use. After performance of the real-life working tasks, this correlation was found to be moderate. It was also observed that the ambient environmental outdoor conditions and personal protective clothing influenced the accuracy and resulted in unrealistic ear canal temperature outliers. It was found that the Cosinuss° °Temp thermometer did not result in significant interference during work. Therefore, it was concluded that, without a correction factor, the Cosinuss° °Temp thermometer is inaccurate. Nevertheless, with a correction factor, the reliability of this wearable ear canal thermometer was confirmed at rest, but not in outdoor working conditions or while wearing a helmet or hearing protection equipment.

## 1. Introduction

Heat stress is an important factor that should be considered in physically demanding occupations. Heat stress is generally influenced by many factors, such as heavy workload, use of personal protective equipment (PPE), and environmental factors (e.g., sun exposures, heat, humidity), and is of major concern among workers with physically demanding occupations [1,2,3,4]. Next to metabolic heat production from the physical work, the two main factors influencing heat stress are the environmental conditions and use of personal protective clothing (PPC) and PPE [1,5]. Working in hot (sunny and humid) environments causes the body temperature to increase as a result of the inhibition of body heat loss [1,3,4,6,7,8]. Wearing full-body PPC and PPE hinders heat removal and so stimulates and increases heat stress as a result of thermal insulation and evaporative resistance [1,2,9,10]. Both heat stress and the strain resulting from it are influenced by individual factors [7,11,12], such as age, health, and fitness level [9,13], and may lead to subsequent health problems, such as exhaustion, dehydration, mental confusion, and loss of consciousness [8,10,14]. This may affect productivity and risk perception and may cause safety problems [10,15,16,17]. In extreme cases, heat strain may cause permanent damage and may even be life-threatening [5,11] and repeated or prolonged heat stress may even lead to cardiovascular disease [18].

Monitoring the workers’ body (core) temperature and the ambient working conditions may potentially help prevent heat strain [19]. Body (core) temperature can be measured using several invasive and non-invasive methods [20]. Although invasive measurements, such as esophageal, rectal, and gastrointestinal thermometers, are highly reliable [21,22,23], their application is not suitable in daily work situations [9,24,25,26]. Moreover, although non-invasive methods, such as ear, skin, and forehead thermometry, have become wearable, they are often impractical in work situations because they either interfere with the working conditions [27] or are unreliable at the individual level [17,22,25,28,29]. Therefore, up till now, no accurate instruments are available for continuously and non-obtrusively monitoring heat strain while performing physically demanding work [15,30,31,32]. Therefore, there is a need for a reliable, non-invasive, continuous temperature measuring system in the form of a wearable thermometer to be able to perform real-time monitoring and prevent heat strain in physically active individuals [15,19,23].

Cosinuss° °Temp (Cosinuss° GmbH, Munich, Germany) is a new, non-invasive, wearable thermometer that can continuously measure ear canal temperature in real time. Research on the accuracy of this thermometer has shown a systematic difference of −1.5 °C compared to infrared (IR) tympanic temperature [30,33]. Such a systematic difference can be compensated for by making software adjustments to the measuring device. Although this thermometer appeared to accurately measure core temperature at rest, its accuracy appeared to be low during firefighting tasks [33], which may be due to some factors as follows. First, the ear canal temperature may not be representative of the deep body temperature while performing physically demanding activities [20,34,35]. Second, imprecise alignment in the ear may result in measuring the aural temperature instead of the ear canal temperature [33,34,36,37]. Third, if the insulation of the thermometer is insufficient in the ear [1,5,6], it is possible for the environmental or local temperature to influence the measurements [23,34]. To summarize, the Cosinuss° °Temp thermometer may form the basis for a non-invasive, non-obstructive monitoring system for workers with physically demanding (outdoor) occupations. Therefore, it is necessary to investigate its accuracy in more detail to explore the factors influencing it, to assess whether its accuracy can be improved or whether it is possible to prevent its accuracy from decreasing, and to determine the type of work or application in which this system may potentially be used, including different working conditions, such as hot and humid factories and outdoor working conditions.

In this study the in vitro and in vivo accuracy of the Cosinuss° °Temp thermometer as a wearable thermometer used for monitoring heat stress among workers with physically demanding occupations with different working conditions was investigated. The aims were to (1) test the in vitro accuracy of the Cosinuss° °Temp thermometer under controlled laboratory conditions; (2) test the in vivo accuracy of the Cosinuss° °Temp thermometer as an ear canal thermometer under controlled laboratory conditions; (3) test the in vivo accuracy of the Cosinuss° °Temp thermometer while performing physically demanding work; (4) investigate the influence of environmental conditions (e.g., wind, temperature changes, and lack of ventilation resulting from wearing PPC or PPE) on the accuracy of the Cosinuss° °Temp thermometer; and (5) explore the usability of the Cosinuss° °Temp thermometer for measuring the ear canal temperature while performing physically demanding work.

## 2. Materials and Methods

### 2.1. Subjects

The inclusion criterion was that the subjects should be physically active workers between 18 and 67 years of age (representing the European working population). The exclusion criteria included lung diseases, cardiovascular disease, claustrophobia, and problems associated with body heat loss (e.g., heat intolerance or difficulties with body thermoregulation resulting from sweating problems). The minimum sample size was calculated using a power analysis (non-inferiority trial with a power of 95%, significance level of 0.05, and acceptable difference of ±0.2 °C) for the laboratory study on the basis of the expected outcomes (*n* ≥ 11 subjects) and for the field study on the basis of the results of the laboratory study (*n* ≥ 26 subjects, with *n* ≥ 7 per job category).

The subjects of the laboratory (control) study were 15 volunteers (mean age: 25.1±4.2 years, nine males, six females) with no experience of wearing PPC or PPE, whereas the subjects of the field study were 49 physically active workers (mean age: 40.4 ± 10.2 years, 47 males, two females). All subjects were recruited by distributing flyers in selected companies with different working situations: (1) chemical cleaners working with chemical-proof PPC combined with other PPE; (2) mechanics working in a warm, humid factory; (3) firefighters working with PPC and PPE; and (4) neighborhood maintenance workers working outdoors in different weather conditions. The diversity in work-related tasks and working conditions between the subject groups, as specified, provided a broad picture of their influence on the accuracy and usability of the discussed wearable thermometer.

This study was performed in accordance with the Code of Ethics of the World Medical Association (Declaration of Helsinki) for experiments involving humans. This study was approved by the Medical Ethics Committee of University Medical Center Groningen, the Netherlands, stating that it does not involve medical research under Dutch law (laboratory study: M16.197311, field study: M17.209969). All subjects were informed of the study via an information letter and a verbal explanation before the start of the study, and all of them signed an informed consent form before participating in the study.

### 2.2. Materials

#### 2.2.1. Cosinuss° °Temp

Cosinuss° °Temp (Cosinuss° GmbH) is an ear canal thermometer that can be worn in and around the ear like a hearing aid (dimensions: 45 × 38 × 18 mm, weight: 6.5 g), as shown in Figure 1. Temperature is measured using a thermistor contact sensor integrated into a sensor head, which is placed in the ear canal. Then, data are sent via Bluetooth Smart 4.0 and viewed on the Cosinuss° One smartphone application. According to the specifications, the accuracy of the Cosinuss° °Temp thermometer is ±0.1 °C, with a measurement range of 0 °C to 50 °C and a working temperature range of −15 °C to 55 °C [38].

#### 2.2.2. Ambient Conditions Box

The ambient conditions box, worn with elastic chest belts (see Figure 2), contains a temperature and humidity sensor (SHT15 Breakout; Sensirion, Stäfa, Switzerland) mounted on the outside of the box. This box is worn under PPC and PPE to measure the micro-climate around the skin of the subject under their clothes (described as the temperature inside clothing, T_cli_), as well as relative humidity (RH). This T_cli_ sensor has a measurement range of −40 °C to 120 °C and an accuracy of ±0.3 °C at 25 °C [39]. It can also measure the RH with an accuracy of ±2% at 10% to 90% with a humidity range of 0% to 100% and a response time of 5–20 s [39]. This box was validated in a climatic test cabinet (type C-40/350; CTS Clima Temperatur Systeme GmbH, Hechingen, Germany) with Pt100 thermometers with an accuracy of T ± 0.3 °C and RH ± 1.5% [40], resulting in a high to very high correlation compared to the climatic test cabinet and Pt100 thermometers.

#### 2.2.3. Reference Ear Canal Infrared Thermometer

The Braun ThermoScan^®^ 7 IRT6520 ear canal IR thermometer (Braun GmbH, Kronberg, Germany) has an accuracy of ±0.2 °C and can perform measurements within the temperature range of 35 °C–42 °C (RH 10–95%) [41]. This thermometer is a commercially available ear canal measurement device for hospital use that was calibrated according to the national standard for medical devices (EN ISO 14971:2012 and EN ISO 10993-1:2009), electrical equipment (EN 60601-1:2006 and 2007, EN 60601-1-11:2010 14971), and clinical thermometers (EN 12470-5:2203) [41] by an authorized service center. All measurements with this thermometer were performed in offices with a constant room temperature of 20.0 ± 2.0 °C and 45.0 ± 5.0% humidity.

#### 2.2.4. Mercury Thermometer

As a reference, a mercury thermometer was used (ET 31; Lauda Dr. R. Wobser GmbH & Co., Lauda-Königshofen, Germany). This mercury-in-glass thermometer has a measurement range of 0 °C to 100 °C, an immersion depth of 90 mm, and a median thread temperature of 30 °C [42]. The accuracy of this thermometer was calibrated in ice and boiling water with an accuracy of ±0.1 °C according to the national standard for thermometers (EN ISO 14971:2012, EN ISO 10993-1:2009, and EN 12470-5:2203).

### 2.3. Study Design

The in vitro accuracy of the Cosinuss° °Temp thermometer (aim 1) was examined in a thermostatic water bath. At a constant temperature, the temperature of the water bath was first measured with the Cosinuss° °Temp thermometer and a reference ear canal IR thermometer and compared to the results of the reference mercury thermometer. Then, the water temperature was increased from 35 °C to 41 °C in increments of 0.5 °C, and the temperature measured was checked against that of the mercury thermometer. Three measurements were performed at every step, with a frequency of one measurement per minute. The sensors were pointed downward in the water, with an angle of 90° between the sensor’s tip and the water surface. Before starting this laboratory study, the ear canal IR and mercury thermometers were (re)calibrated according to the national standard for medical thermometers (EN ISO 14971:2012, EN ISO 10993-1:2009, and EN 12470-5:2203).

To test the in vivo accuracy of the thermometer under controlled laboratory conditions (aim 2), the ear canal temperature (T_EC_) of the subjects was measured at rest using the Cosinuss° °Temp thermometer and compared to the results obtained from the tympanic IR thermometer. The T_EC_ value was measured 10 times per subject with a frequency of one measurement per minute, yielding a 10 min measurement. All measurements were performed in offices with a constant room temperature of 20.0 ± 2.0 °C and 45.0 ± 5.0% humidity, and all subjects were allowed to acclimatize to this environment (for about 10 min; if necessary, every subject was allowed to have more time with a maximum of 15 min depending on their personal preference or if the T_EC_ value was not stable).

The field study was comprised of three stages: (1) accuracy measurements; (2) performance of daily jobs; and (3) accuracy measurements. To test the in vivo accuracy of the thermometer (aim 3), at stages 1 and 3, the T_EC_ value of the subjects was measured at rest using the Cosinuss° °Temp thermometer and a tympanic IR thermometer. In total, five measurements of T_EC_ were performed, with a frequency of one measurement per minute, yielding a 5-min measurement. In both the laboratory and field studies, ear canal thermometer positioning was monitored continuously and adjusted if needed.

To investigate the influence of real-life working conditions on the accuracy of the Cosinuss° °Temp thermometer (aim 4), during stage 2, the T_EC_ values and environmental conditions of the subjects were monitored while they were performing physically demanding work. All subjects performed their daily jobs while wearing the Cosinuss° °Temp thermometer, and the T_EC_, T_cli_, and RH values were continuously monitored using a wearable ambient conditions box. The duration of stage two depended on the duration of the subject’s task, lasting between 30 min and 3 h. When unrealistically high or low T_EC_ values were observed, the subjects were asked how they feel and whether they have any thermoregulation-related issues or health complaints.

During in vivo accuracy tests of aims 2, 3 and 4, the data gathered with the Cosinuss° °Temp were sent continuously to the ambient condition box via Bluetooth and stored every 1 s.

The usability of the Cosinuss° °Temp thermometer was explored (aim 5) using the AEIOU (Activities, Environments, Interactions, Objects, and Users) user interface design method via researchers’ observations and feedback from the subjects. In this descriptive observational study, the usability aspects were ease of use, positioning, wearability by all types of users, fixation, and comfort. In the laboratory study, subjects (Users) were asked to wear the Cosinuss° °Temp thermometer (Objects) while putting on and removing PPC (TRELLCHEM^®^ chemical-proof hazmat suit, Super type T; Ansell Protective Solutions AB, Trelleborg, Sweden) [43] with a separate gas mask during rest (3 min), while sitting (3 min), while walking (2 min), and while jumping (2 min) in PPC (Activities) for a total of 10 min. These two tests were performed directly after each other under constant ambient conditions (T_a_ = 20.0 ± 2.0 °C, RH = 45.0 ± 0.5%) (Environments). Under real-life working conditions (Environments), this was equivalent to the performance of daily jobs (Activities) in physically demanding occupations (Users).

### 2.4. Data Analysis

To check whether the in vivo difference in the accuracy of the Cosinuss° °Temp thermometer compared to the IR thermometer was due to a misalignment in the ear canal caused by the thermometer’s positioning in the ear and/or individual differences (inner-ear dimensions), a correction factor was introduced. This individual correction factor was calculated using the second accuracy measurement (randomly chosen from the first five measurements out of 10 in the laboratory study and from the first three measurements out of five in the field study) during the in vivo measurements at rest. The T_EC_ value was measured using an IR thermometer in combination with a Cosinuss° °Temp thermometer in the other ear, and the difference between their measurements was used as the correction factor.

All statistical analyses were performed using IBM SPSS Statistics (version 25; IBM Corp., Armonk, NY, USA). To test the in vitro accuracy of the thermometer (aim 1), the mean of three measurements per step was used in the laboratory study. To statistically analyze the in vivo accuracy of the Cosinuss° °Temp thermometer (aim 2), every ninth measurement (out of 10) was used in the laboratory study and every fourth measurement (out of five) was used in the field study (aim 3). Differences were assessed using a paired *t*-test, and an intraclass correlation coefficient (ICC, two-way mixed model, absolute agreement) was calculated for normally distributed data. The ICC was considered low at <0.39, moderate at 0.40–0.59, high at 0.60–0.79, and very high at ≥0.80 [44]. Non-parametric data were also tested using Wilcoxon’s signed rank test. All *p*-values less than 0.05 were considered statistically significant. The limit of agreement (LoA) reflects the average difference between two different measurements and is calculated as ±1.96*SD difference [45]. For this study, the acceptable level for accuracy was set at mean difference (MD) ± 0.2 °C, with a moderate or (very) high ICC (≥0.40) and an LoA value of ≤0.50. Bland–Altman plots were created to analyze the individual differences between measurements against the individual mean of the two measurements. Generally, a funnel shape indicates that the magnitude of the difference is related to the mean performance. Sensitivity analyses were also performed to test the differences between the ninth and tenth measurements (laboratory study) and between the fourth and fifth measurements (field study), as well as for all the measurements. Descriptive statistics were used to analyze the usability in the laboratory and field studies (aims 3 and 5).

## 3. Results

### 3.1. In-Vitro Accuracy

It was observed that the mean temperature difference in the thermostatic water bath between the mercury thermometer and the Cosinuss° °Temp thermometer was −0.4 ± 0.2 °C (MD ± SD; *p* < 0.001), whereas that between the mercury thermometer and IR thermometer was −0.2 ± 0.1 °C (*p* < 0.001). Table 1 shows the results of the ICC analysis for the three thermometers.

Table 1 shows a very high correlation between the Cosinuss° °Temp and IR thermometers compared to the mercury thermometer (ICC ≥ 0.97), with an LoA value of ≤0.37 (within the acceptable level of 0.50). Figure 3 shows Bland–Altman plots. Sensitivity analysis revealed non-significant differences.

### 3.2. In-Vivo Accuracy under Controlled Conditions

It was observed that the mean T_EC_ value measured using the Cosinuss° °Temp thermometer was 35.2 ± 0.6 °C with a within-subject difference of 0.1 ± 0.1 °C. Table 2 shows the mean differences and results of the ICC analysis.

It was observed that the mean T_EC_ difference between the Cosinuss° °Temp thermometer and the IR thermometer exceeded the acceptable level (MD = −1.4 °C, *p* < 0.001) and exhibited a low correlation (ICC = 0.07, *p* = 0.083). However, after an individual correction factor (mean: 1.4 ± 0.6 °C, min: 0.7 °C, max: −2.5 °C) was applied, this mean difference decreased to an acceptable level (0.0 °C, *p* = 0.729) and a high correlation (ICC ≥ 0.72) was found. Figure 4 shows a Bland–Altman plot. Without correction, the Bland–Altmann plot showed bias between the mean differences; however, no funnel shape was visible. Sensitivity analysis revealed similar ICC and *p*-values.

### 3.3. In Vivo Accuracy under Real-Life Conditions

In five subjects of the field study, the Cosinuss° °Temp thermometer stopped working because of the sweat that they produced while performing their jobs; the system was not fully waterproof, resulting in sweat on the electrical components (circuit board), corroding them over the course of the study. This resulted in loss of data and fewer results after work. Despite the incomplete datasets, all subjects were included in the accuracy study.

It was found that the mean T_EC_ of the Cosinuss° °Temp thermometer during the accuracy measurements was 35.0 ± 1.4 °C (mean ± SD) without correction. The mean correction factor was −1.7 ± 1.1 °C (min: 0.1 °C, max: −6.0 °C, outlier). Table 3 shows the mean differences and results of the ICC analysis for raw and corrected systems before and after work.

Before correction, an unacceptable high difference (MD = 1.5 °C, *p* < 0.001) and a low correlation (ICC ≤ 0.25, *p* ≤ 0.021) were observed, consistent with the laboratory study. However, after correction, the mean difference decreased to an acceptable level (MD = −0.2 °C, *p* ≤ 0.110) and the correlation between the Cosinuss° °Temp thermometer and IR thermometer became high. Before working an acceptable LoA value was observed (ICC = 0.77, *p* < 0.001, LoA = ±0.43). After the subjects performed their jobs, it was observed that the correlation between the Cosinuss° °Temp thermometer and IR thermometer decreased to a moderate level, exceeding the acceptable level of the LoA (ICC = 0.55, *p* < 0.001, LoA = ±1.90). Figure 5 shows Bland–Altman plots. Sensitivity analysis revealed similar results, with the sensitivity analysis of complete datasets only revealing non-significant differences.

### 3.4. Influences of Real-Life Working Conditions on Accuracy

While the workers were performing their jobs, it was observed that the mean T_EC_ was 36.8 ± 1.6 °C, with a mean T_cli_ value of 26.9 ± 4.9 °C and mean RH of 62.6 ± 12.7%. Table 4 shows the mean T_EC_, T_cli_, and RH measured using the Cosinuss° °Temp thermometer while the workers were performing their jobs. Each subject exhibited individual patterns of development of T_EC_, with the micro-climate (T_cli_) and RH differing from one job to another. Figure 6 shows the values of T_EC_, T_cli_, and RH over time while four representative individuals were performing their jobs. Table 4 andTable 5 outline the mean differences and results of the ICC analysis for the four different jobs mentioned earlier (with corrected Cosinuss° °Temp thermometer data; the raw Cosinuss° °Temp thermometer data are shown in Appendix B).

In the case of mechanics and neighborhood maintenance workers, it was observed that the working environment (i.e., temperature, RH) played a major role and influenced the development of T_EC_, T_cli_, and RH while they were performing their jobs. For mechanics, the values of T_cli_ and RH in the indoor working environment were constant within the subjects (see Figure 6, mechanics), resulting in a relatively constant T_EC_. Before and after work, the mean difference was within the acceptable level (MD ≤ ±0.2, *p* ≥ 0.087) with a (very) high correlation (ICC ≥ 0.63, *p* ≤ 0.005). For neighborhood maintenance workers, it was observed that the outdoor working environment fluctuated, resulting in small fluctuations in T_EC_ (see Figure 6, neighborhood maintenance worker). Although the mean difference was within the acceptable level (MD ≤ ±0.2, *p* ≤ 0.796), the ICC value decreased from high (ICC = 0.67, *p* = 0.001) to negative, indicating a non-random effect influenced by a third variable [46]. For two mechanics and four neighborhood maintenance workers, the value of T_EC_ was lower than 35.0 °C. Given these results, the six subjects were asked how they felt, and all of them mentioned that they did not have any thermoregulation-related or health complaints. All six of them worked in cold, rainy/foggy, and/or windy environments, which implies that these measurement errors may have been due to the environmental conditions (e.g., wind or temperature of the working environment) [46].

In the case of chemical cleaners and firefighters, both the PPE (breathing apparatus) and chemical-proof PPC as well as the fire-proof PPC and helmet caused an increase in T_cli_, RH, and T_EC_ as a result of transpiration and lack of ventilation. It was observed that when the PPC was removed after finishing the task, the values of T_cli_ and RH dropped rapidly, followed by the T_EC_ value (see Figure 6, firefighters). Before work, the firefighters showed a moderate correlation (ICC = 0.51, *p* < 0.001), which decreased to a low correlation after work (ICC = 0.28, *p* = 0.110). For chemical cleaners, this correlation remained high (ICC ≥ 0.60, *p* ≤ 0.029). While working, the T_EC_ value of nine subjects exceeded 40.0 °C, with abnormal values reaching a maximum T_EC_ of 46.4 ± 2.0 °C. These nine subjects comprised three chemical cleaners wearing PPC and PPE, four firefighters wearing PPC and PPE, and two neighborhood maintenance workers wearing temporary PPE (hearing protection). None of these subjects had any thermoregulation-related or health complaints, which implies that such an abnormal increase was due to them wearing PPC and PPE or due to environmental conditions. Sensitivity analysis (of complete datasets) revealed similar results or non-significant differences, and similar patterns were observed in all job types.

### 3.5. Usability

Generally, the Cosinuss° °Temp thermometer is easy to position in and around the ear. All subjects experienced a good fit with the Cosinuss° °Temp thermometer, and most of them reported it to be comfortable and to look professional. In the laboratory study, at rest, two subjects reported that the Cosinuss° °Temp thermometer felt loose and that they needed to reposition it by pushing it a little into their ears. It was reported that the thermometer remained in place during sitting, walking, and jumping in PPC. However, in all cases (*n* = 15), while the subjects were either wearing or taking off their chemical-proof suits, the thermometer fell out of their ears.

In real-life working conditions, all subjects reported that the Cosinuss° °Temp thermometer was non-obstructive and in general remained well fixated within their ears while they were performing their jobs. In two subjects, the thermometer fell out of the subjects’ ears while they were performing their jobs. In three subjects, the Cosinuss° °Temp thermometer felt loosely fixated within the ear and was repositioned by the subject. In eight subjects (seven chemical cleaners, one firefighter), the thermometer fell out of their ears while they were taking off their PPC, helmets, or head-related parts of their chemical-proof suits. Figure 6 shows how the falling of the thermometer out of the ear reflects in a drop of T_EC_ (see Figure 6, chemical cleaners). Importantly, the Cosinuss° °Temp thermometer did not interfere with task performance. Although the firefighters mentioned that there may be some complications for workers who need to wear in-ear hearing protection equipment or an in-ear communication device.

## 4. Discussion

The aim of this study is to explore and evaluate the accuracy and reliability of the Cosinuss° °Temp thermometer as a wearable thermometer used in monitoring the ear canal temperature while performing physically demanding work. For hospital use, both the Cosinuss° °Temp thermometer and IR thermometers showed very high correlations compared to mercury thermometers in an in vitro study. However, a laboratory and field study indicated the need for a correction factor and calibration before use, because the Cosinuss° °Temp thermometer showed a continuously lower ear canal temperature of −1.4 °C, which is in line with previous research [33], as well as research on Cosinuss° One [30]. Monitoring the in-ear placement of the thermometer and applying an individual correction factor caused the accuracy of the Cosinuss° °Temp thermometer to increase, indicating the influence of individual ear/in-ear dimensions and lack of fit in the ear. After correction, the results obtained in the in vivo studies supported the reliability of this system. It should be noted that, on the one hand, obtaining unrealistically low temperatures indicates the influence of weather conditions (e.g., wind) on the accuracy and reliability of the Cosinuss° °Temp thermometer and the need for insulation. On the other hand, obtaining unrealistically high temperatures indicates the influence of PPC and PPE on the accuracy of the Cosinuss° °Temp thermometer. In general, lack of ventilation and/or air circulation while wearing PPC and PPE around the ear results in local warming of the environment near the ear. Both of these findings indicate that the Cosinuss° °Temp thermometer is sensitive for measuring environmental conditions instead of ear canal temperature. To check for possible systematic errors, Pearson’s test was used to analyze the mean differences between the Cosinuss° °Temp thermometer and the reference thermometer. A systematic error was detected in the raw results of the Cosinuss° °Temp thermometer, while no error was detected with the ear canal IR thermometer or corrected Cosinuss° °Temp thermometer. Moreover, under some working conditions, the Cosinuss° °Temp thermometer was found to be less accurate then without wearing PPC, PPE or working indoor, likely because of the lack of insulation in the ear while performing physically demanding work or lack of close fit. Therefore, it is important to improve the fit of the Cosinuss° °Temp device.

Compared to currently available research, one of the strengths of this study is that we explored the interaction among the user (human), device, and different working environments. This study was performed both in a laboratory and in real-life working environments. The accuracy of the Cosinuss° °Temp thermometer was assessed in multiple work situations to gain an insight into the influence of PPC and PPE as well as the environmental conditions on the thermometer. In the field study, the subjects were workers who experienced temperature-related challenges while performing their jobs. Because they work under similar conditions on a daily basis, that is, under different (outdoor) environmental conditions from those of neighborhood maintenance workers, they were able to provide adequate feedback. Hence, we were able to gain insights into the development of body temperature among different types of workers, which is relevant for an objective and more accurate prediction of heat strain. This represents a first step in improving the health and safety of workers with physically demanding occupations.

A limitation of this study was the ear canal infrared reference thermometer. Although this non-invasive and fast reference [9] is currently the measurement standard for hospital use [47,48], it can probably not be considered as the gold standard for lab-based studies [23,36,37,49]. Measuring the inner-ear temperature is a reliable method [49,50,51] for monitoring body temperature in scientific research [30,52,53,54], although some studies have shown that this method is less accurate than other methods. This method usually results in less accurate (±1.0 °C) measurements [23,36,37,49], given that the local temperature could be measured instead of the ear canal temperature [35,55,56]. This method can, however, be applied in working conditions [57] in which workers are expected to be subjected to excessive heat strain [26]. Although concurrent validity measurements using rectal thermometers, invasive “temperature pills” (mini thermometers that can be swallowed), or zero-heat-flux thermometers are preferable [23,58], they are impossible to apply in real-life working conditions because they do not work continuously and are either impractical (e.g., rectal thermometer) or are influenced by hot/cold food and liquid intake (temperature pill), which is crucial in physically demanding occupations [20,25,27,31]. One of the limitations in this study was that the Cosinuss° °Temp thermometer could not be validated in the presence of PPC and/or PPE. Although we could not prove that the Cosinuss° °Temp thermometer shifted as a result of movement, this is the most likely reason. Moreover, because the Cosinuss° °Temp thermometer is not explosion-proof, it was not allowed in all working locations, resulting in a limited representation of the working population.

It should be noted that the Cosinuss° °Temp thermometer is sensitive to misalignment in the ear canal, which may lead to the measurement of the aural temperature instead of the tympanic membrane temperature [1,20,35]. This may result in the over- or underestimation of the actual heat strain [23,36,37,49]. Perhaps this can be prevented using individually tailored components; hence, more research in this direction is recommended. Suggestions for further research include studying the properties of the Cosinuss° °Temp thermometer in the laboratory using an invasive scientific standard to investigate its accuracy compared to deep body temperature measurements. Moreover, it is important to investigate the correction factor (before each use of the sensor) and calibration method in more depth in a repeatability study. It is also crucial to further validate this system over a full day and on different working days, as well as in other types of work and environments. Furthermore, researchers should also investigate the development and relationship between ear canal temperature, environmental temperature, and humidity, as well as other physiological parameters, such as the heart rate [17,24], to make full use of this thermometer to prevent overheating on an individual level. Developing a mathematical algorithm may help combine the output of the parameters and increase the accuracy of heat strain prediction. The system also needs to be optimized in terms of accuracy and stability, including its fixation and isolation in the ear, taking into account different ear shapes. In addition, the system needs to be resistant to explosions and to be equipped with more sensors to measure more parameters, such as heat radiation and air velocity. Although none of the subjects reported that the Cosinuss° °Temp thermometer interfered with their workability, the thermometer slightly reduced their hearing capability. For workers who need to wear in-ear hearing protection equipment, further research should be performed to make the ear attachment soundproof to protect the workers against noise or to integrate the device in hearing protection equipment. For workers who use in-ear communication, it is important to make the Cosinuss° °Temp thermometer compatible with or integrated within communication systems. This would be a great advantage for workers such as firefighters.

In general, it is important to develop a continuous, non-invasive instrument that can prevent heat strain among workers and improve their health and safety while working [30,31,32,59]. In general, wearable thermometers should be suitable for all body proportions, reusable with low costs, and resistant to the effects caused by food and liquid intake [20,27,31]. The results obtained in this study show that the Cosinuss° °Temp thermometer can be used to monitor the ear canal temperature over time during work. It can also be used for screening purposes [57], by providing an insight into the risks that may lead to excessive heat strain, and may potentially help prevent work-related overheating and dehydration [32,49] for workers. On the one hand, on the group level, it can be used to gain an insight into the risks that may lead to excessive heat strain. On the other hand, on the individual level, it needs to be more accurate at rest and requires individual calibration. It should be noted that this device can be used in other fields as well, such as health care [60,61], remote health monitoring [59,62,63], sports applications, and even space [64].

## 5. Conclusions

From the results of the laboratory study, high correlations were observed between the Cosinuss° °Temp thermometer and tympanic IR thermometers. Furthermore, an in vivo accuracy study showed the need for individual correction factors for the Cosinuss° °Temp thermometer. Without these corrections, this thermometer lacks accuracy. When a correction factor was applied to correct individual differences, moderate to high correlations were found with an acceptable reliability. At the beginning of physically demanding work, the Cosinuss° °Temp thermometer showed a very high correlation for measuring the ear canal temperature. However, after a certain amount of time, this correlation decreased and became moderate, with some unrealistic ear canal temperature results obtained while the workers were performing their jobs. This was probably due to environmental factors (e.g., wind), wearing helmets and hearing protection equipment, and lack of insulation of the area around the ear. When an individual correction factor was used, it was confirmed that the thermometer is reliable at rest, but not during outdoor work or while wearing helmets and/or hearing protection equipment. Hence, the Cosinuss° °Temp thermometer is comfortable and causes minimal interference (or not at all) during work. It can be concluded that the Cosinuss° °Temp thermometer with a correction factor can be used to monitor the development of the individual ear canal temperature of workers with physically demanding occupations, but not in outdoor working conditions or while wearing a helmet or hearing protection equipment.

## Figures and Tables

**Figure 1 ijerph-18-04896-f001:**
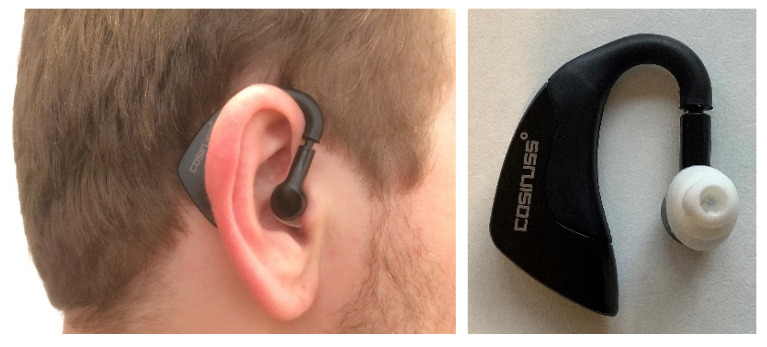
The wearable ear thermometer Cosinuss° °Temp. Left: Cosinuss° position in the ear. Right: Cosinuss° with IR sensor in the white ear tip.

**Figure 2 ijerph-18-04896-f002:**
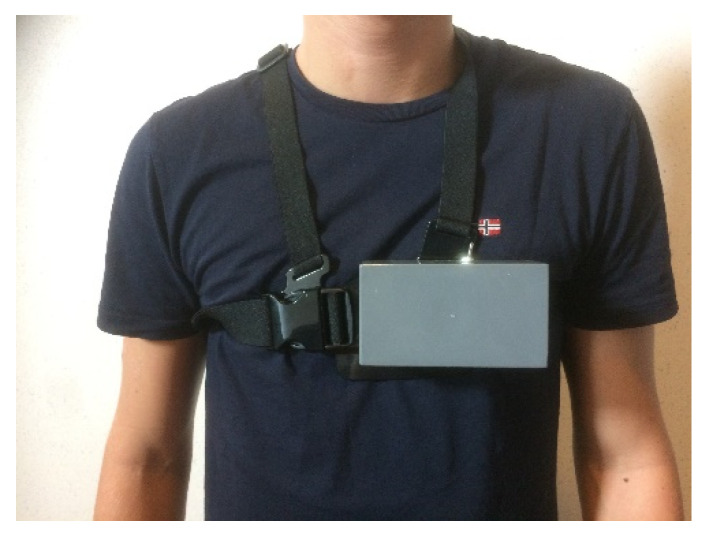
Ambient conditions box with temperature and humidity sensors and data receiver and storage.

**Figure 3 ijerph-18-04896-f003:**
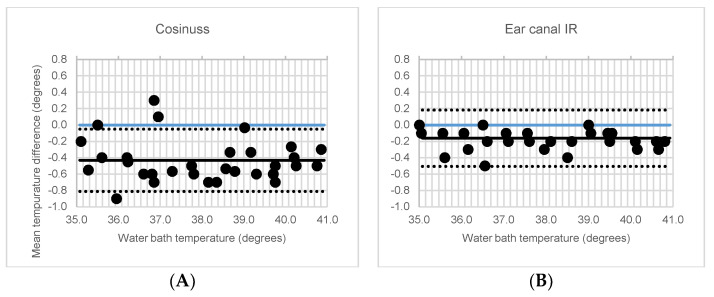
Bland-Altman plots of the mean temperature versus the mean temperature difference. The (**A**) Cosinuss° °Temp, and (**B**) infrared (IR) thermometer compared to mercury thermometer with mean (black), upper and lower limit of agreement (LoA) (black dotted line) and zero-line (blue).

**Figure 4 ijerph-18-04896-f004:**
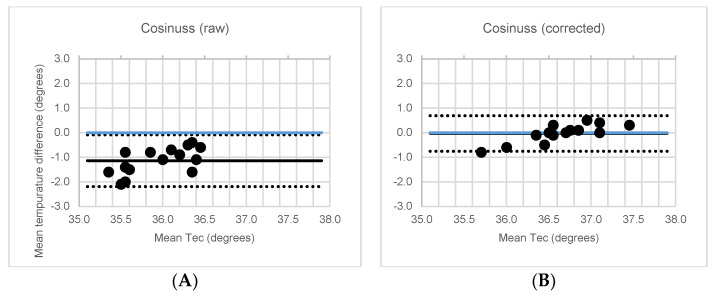
Bland-Altman plots of the mean ear canal temperature versus the mean temperature difference. The ear canal temperature (TEC) measured with non-corrected (**A**) Cosinuss° and corrected (**B**) Cosinuss° compared to the IR thermometer with mean and upper and lower limit of agreement (LoA).

**Figure 5 ijerph-18-04896-f005:**
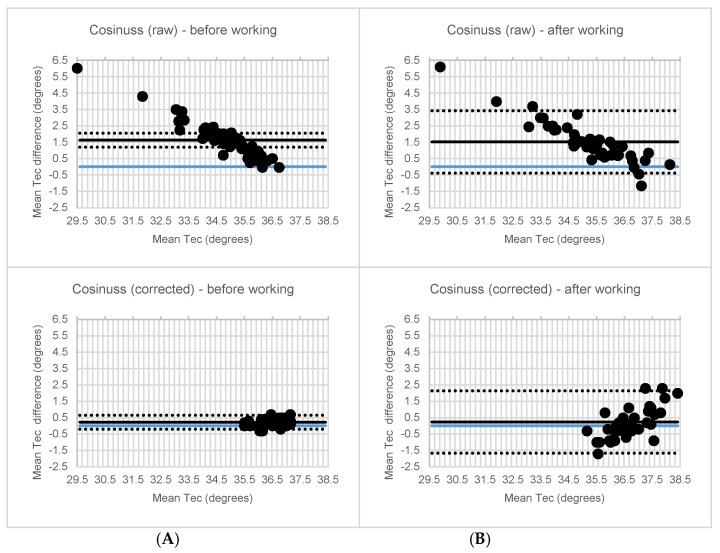
Bland-Altman plots of the mean ear canal temperature versus the mean temperature difference. The ear canal temperature (T_EC_) raw and corrected Cosinuss° (**A**) before and (**B**) after working compared to the IR thermometer with mean and upper and lower limit of agreement (LoA).

**Figure 6 ijerph-18-04896-f006:**
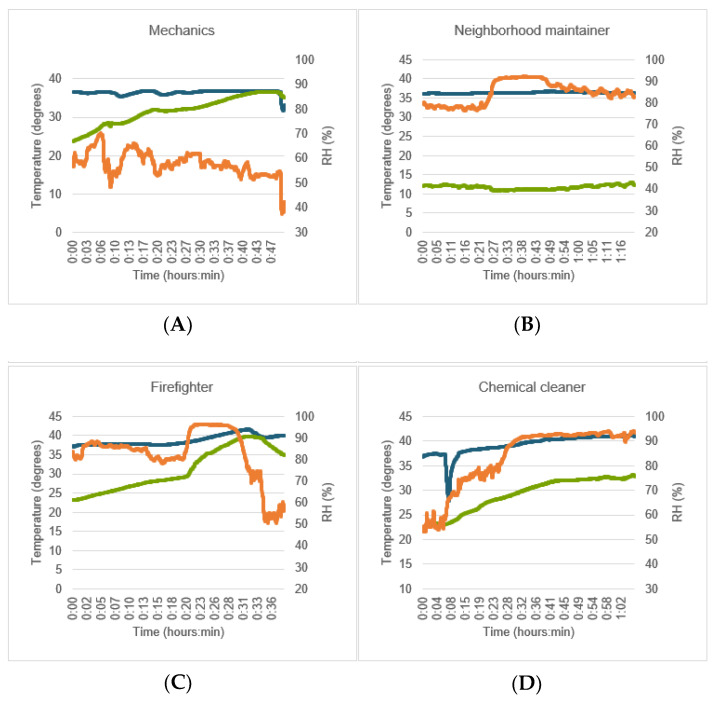
Individual graph during the performance of the job. (**A**) Mechanics; (**B**) neighborhood maintainer; (**C**) firefighters; (**D**) chemical cleaner. Ear canal temperature (°C) (dark blue), nearby micro-climate temperature (°C) (green) and relative humidity (RH) (%) (orange) of 4 subjects with different jobs.

**Table 1 ijerph-18-04896-t001:** Cosinuss° and IR thermometer versus mercury thermometer using the paired t-test with mean difference (MD) with standard deviation (SD) and the intraclass correlation coefficient (ICC) with a confidence interval of 95%, *p*-value and limits of agreement (LoA).

	MD ± SD [95% CI]	*p*	ICC [95% CI]	*p*	LoA
Cosinuss°	−0.4 ± 0.2 [−0.6;−0.3]	<0.001	0.97 [0.13;1.00]	<0.001	±0.37
Ear canal IR	−0.2 ± 0.1 [−0.3;−0.1]	<0.001	0.99 [0.87;1.007]	<0.001	±0.24

**Table 2 ijerph-18-04896-t002:** Cosinuss° versus ear canal IR thermometer. Cosinuss° versus ear canal IR thermometer. The Cosinuss° was compared with the references using the intraclass correlation coefficient (ICC) with a confidence interval of 95%, *p*-value and limits of agreement (LoA) (n = 15).

	MD ± SD [95% CI]	*p*	ICC [95% CI]	*p*	LoA
Raw	1.4 ± 0.5 [1.1;1.7]	<0.001	0.07 [−0.05;0.31]	0.083	±1.05
With correction factor	0.0 ± 0.4 [−0.2;0.2]	0.729	0.72 [0.33;0.90]	0.001	±0.72

**Table 3 ijerph-18-04896-t003:** Cosinuss° non-corrected and corrected per subject. The Cosinuss° was compared with the IR thermometer per company using the intraclass correlation coefficient (ICC) with a confidence interval of 95% and *p*-value.

Working	MD ± SD [95% CI]	*p*	ICC [95% CI]	*p*	LoA
Raw					
Before working (*n* = 49)	1.5 ± 1.2 [1.1;1.8]	<0.001	0.13 [−0.08;0.37]	0.021	±2.23
After working (*n* = 43)	1.5 ± 1.2 [1.1;1.8]	<0.001	0.25 [−0.09;0.54]	0.002	±2.44
Corrected per subject					
Before working (*n* = 49)	−0.2 ± 0.2 [−0.3;−0.2]	<0.001	0.77 [0.19;0.91]	<0.001	±0.43
After working (*n* = 43)	−0.2 ± 1.0 [−0.5;0.1]	0.110	0.55 [0.32;0.73]	<0.001	±1.90

**Table 4 ijerph-18-04896-t004:** Ear canal temperature, micro-climate temperature and relative humidity of all subjects and per job measured. Mean and max ear canal temperature (T_EC_) (°C), nearby micro-climate temperature (T_cli_) (°C) and relative humidity (RH) (%) of all subjects and per job measured with Cosinuss° and ambient conditions chest box.

Job Type	*n*	Mean T_EC_ (°C) (Corrected)	Mean T_cli_ (°C)	Mean RH (%)	Max T_EC_ (°C)	Max T_cli_ (°C)	Max RH (%)
All subjects	49	36.8 ± 1.6	26.9 ± 4.9	62.6 ± 12.7	46.4 ± 2.0	39.5 ± 6.1	92.6 ± 13.6
Chemical cleaners	9	37.6 ± 1.5	26.9 ± 1.9	78.8 ± 5.7	46.4 ± 2.8	33.0 ± 1.8	91.9 ± 2.1
Mechanics	13	36.1 ± 0.8	28.0 ± 2.2	49.7 ± 9.0	38.3 ± 0.6	36.5 ± 2.7	75.9 ± 10.9
Firefighters	14	37.9 ± 0.7	31.2 ± 2.2	67.6 ± 7.7	41.6 ± 1.2	39.5 ± 3.0	92.5 ± 3.4
Neighborhood maintainers	13	36.0 ± 2.0	21.2 ± 4.8	60.1 ± 8.7	42.7 ± 2.0	28.9 ± 4.8	88.4 ± 8.2

**Table 5 ijerph-18-04896-t005:** Cosinuss° corrected per subject. The Cosinuss° was compared with the ear canal IR thermometer per company using the intraclass correlation coefficient (ICC) with a confidence interval of 95% and *p*-value.

Job Type	*n*	MD ± SD [95% CI]	*p*	ICC [95% CI]	*p*
Before performance of the job					
Chemical cleaners	9	−0.2 ± 0.2 [−0.4;−0.1]	0.007	0.88 [0.16;0.98]	<0.001
Mechanics	13	−0.1 ± 0.2 [−0.3;0.0]	0.087	0.81 [0.47;0.94]	<0.001
Firefighters	14	−0.3 ± 0.2 [−0.4;−0.3]	<0.001	0.51 [−0.08;0.85]	<0.001
Neighborhood maintainers	13	−0.2 ± 0.3 [−0.4;−0.1]	0.012	0.67 [0.11;0.90]	0.001
After performance of the job					
Chemical cleaners	7	−0.4 ± 0.6 [−0.9;0.1]	0.083	0.60 [−0.06;0.91]	0.029
Mechanics	13	0.2 ± 0.5 [−0.1;0.5]	0.121	0.63 [0.18;0.87]	0.005
Firefighters	13	−0.7 ± 1.0 [0.6;1.9]	0.029	0.28 [−0.16;0.68]	0.110
Neighborhood maintainers	10	0.1 ± 1.2 [−0.8;1.0]	0.796	−0.09 [−0.77;0.57]	0.594

## Data Availability

The authors confirm that the data supporting the findings of this study are available within the article and its Appendix A.

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
