# Peer review of "Evaluation of a Wearable Non-Invasive Thermometer for Monitoring Ear Canal Temperature during Physically Demanding (Outdoor) Work"

_ijerph, 2021, doi:10.3390/ijerph18094896_

Round 1

Reviewer 1 Report

The manuscript provides detailed description of experimental investigations and analysis of obtained results. It was very interesting study and well done report. However, please consider some minor improvements:

1) In the section Results – the accuracy of data provided in tables and in their description should be unified (e.g. -0.4 and -0.44), as well as terms and symbols (e.g. mean and MD). It is also suggested to explain all terms and symbols used in tables (e.g. Before and After used in table 3);  

2) Please check consistency between Table 3 and description in lines 301-310;  

3) Please provide characteristic of the use of Bluetooth communication between thermometer used in experiments – is it active during the measurement sessions or only after the session when measurement results are stored (or in other way – is the thermometer fully autonomous during the measurement sessions or managed via Bluetooth wireless links) – It is important because of potential limitations of the use of measurement tools in the work environment exposed to strong electromagnetic radiation, as well as because of possible influence on the wireless communication caused by the use of PPE and PPC, and also exposure of worker caused by Bluetooth emitters 

I would like also to wish authors successful developing of further investigations.

Author Response

The manuscript provides detailed description of experimental investigations and analysis of obtained results. It was very interesting study and well done report. However, please consider some minor improvements:

  • We thank the reviewer for this positive comment.

1) In the section Results – the accuracy of data provided in tables and in their description should be unified (e.g. -0.4 and -0.44), as well as terms and symbols (e.g. mean and MD). It is also suggested to explain all terms and symbols used in tables (e.g. Before and After used in table 3);  

  • Adjusted throughout the manuscript. In the tables one decimal has been used in line with the text. The terms and symbols have been unified. “Before” and “After” has been unified to “before working” and “after working”.

2) Please check consistency between Table 3 and description in lines 301-310;  

  • Checked and improved (lines 304-312): “Before correction, an unacceptable high difference (MD=1.5, p<0.001) and a low correlation (ICC≤0.25, p≤0.021) were observed, consistent with the laboratory study. However, after correction, the mean difference decreased to an acceptable level (MD=−0.2, p≤0.110) and the correlation between the Cosinuss° °Temp thermometer and IR thermometer became high. Before working an acceptable LoA value was observed (ICC=0.77, p<0.001, LoA=±0.43). After the subjects performed their jobs, it was observed that the correlation between the Cosinuss° °Temp thermometer and IR thermometer decreased to a moderate level, exceeding the acceptable level of the LoA (ICC=0.55, p<0.001, LoA=±1.90).”

3) Please provide characteristic of the use of Bluetooth communication between thermometer used in experiments – is it active during the measurement sessions or only after the session when measurement results are stored (or in other way – is the thermometer fully autonomous during the measurement sessions or managed via Bluetooth wireless links) – It is important because of potential limitations of the use of measurement tools in the work environment exposed to strong electromagnetic radiation, as well as because of possible influence on the wireless communication caused by the use of PPE and PPC, and also exposure of worker caused by Bluetooth emitters 

  • This is indeed a potential limitation. To prevent these issues, in this study there has been chosen to continuously send and store the data of the wearable ear thermometer to the ambient conditions worn on the chest. With this solution the data did not needed to be send to a mobile phone with the Cosinuss application. Added to study design (lines 202-204): “During in vivo accuracy tests of aim 2, 3 and 4, the data gathered with the Cosinuss° °Temp was send continuously to the ambient condition box via Bluetooth and stored every 1 second.
  • There was not interference between the wireless communication systems used by the workers (firefighters and chemical cleaners) and the wireless Bluetooth connection of the instruments used in this study. However, this could be the case with strong electromagnetic radiations or other wireless communication systems. As mentioned in the discussion: “For workers who use in-ear communication, it is important to make the Cosinuss° °Temp thermometer compatible with or integrated within communication systems. This would be a great advantage for workers such as firefighters.”

Reviewer 2 Report

Overall a nicely undertaken study on an important issue. I have a couple of comments for your consideration:

  1. Abstract, line 19: "Under real life working conditions after work" - I didn't understand "after work" in the context of this sentence?
  2. Introduction, line 35: I didn't understand "next to" in this sentence?
  3. Results, 3.3, line 289: it says that the unit is "not fully waterproof" and this lead to damage of instrument and loss of data. If it is not "waterproof" how could it be tested in a "water bath" in the in-vitro test? Wouldn't the same issues occur in this test if it was in water?
  4. Conclusions, lines 511 to 514: this last concluding sentence seems to  not be supported by the previous sentences? It would seem that with some 'qualifications' this conclusion could be made, but these 'qualifications' are not listed in the concluding sentence.

Author Response

Overall a nicely undertaken study on an important issue.

  • We thank the reviewer for this positive comment.

I have a couple of comments for your consideration:

(1) Abstract, line 19: "Under real life working conditions after work" - I didn't understand "after work" in the context of this sentence?

  • Adjusted (line 19-20): “After performance of the real-life working tasks, this correlation was found to be moderate.”

(2) Introduction, line 35: I didn't understand "next to" in this sentence?

  • Adjusted (lines 35-38): “Next to metabolic heat production from the physical work, the two main factors influencing heat stress are the environmental conditions and use of personal protective clothing (PPC) and PPE [1,5].”

(3) Results, 3.3, line 289: it says that the unit is "not fully waterproof" and this lead to damage of instrument and loss of data. If it is not "waterproof" how could it be tested in a "water bath" in the in-vitro test? Wouldn't the same issues occur in this test if it was in water?

  • In the in-vitro water bath experiment the instrument was not completely under water; the charging input was used to suspend the instrument in the water bath, but this part was not under the water. Additionally, the water bath experiment was performed before the other measurements. while during this experiment water might have entered the instrument, it did not cause errors because it takes time to corrode the electronical components.

(4) Conclusions, lines 511 to 514: this last concluding sentence seems to  not be supported by the previous sentences? It would seem that with some 'qualifications' this conclusion could be made, but these 'qualifications' are not listed in the concluding sentence.

  • Adjusted (lines 515-520): “When an individual correction factor was used, it was confirmed that the thermometer is reliable at rest, but not during outdoor work or while wearing helmets and/or hearing protection equipment. Hence, the Cosinuss° °Temp thermometer is comfortable and causes minimal interference (or not at all) during work. It can be concluded that the Cosinuss° °Temp thermometer with a correction factor can be used to monitor the development of the individual ear canal temperature of workers with physically demanding occupations, but not in outdoor working conditions or while wearing a helmet or hearing protection equipment.”
